# `RecXplainer`: Post-Hoc Attribute-Based Explanations for Recommender Systems

**Sahil Verma**
University of Washington, Seattle
Seattle, WA, USA
`vsahil@cs.washington.edu`

**Anurag Beniwal**
Amazon
`beanurag@amazon.com`

**Narayanan Sadagopan**
Amazon
`sdgpn@amazon.com`

**Arjun Seshadri**
Amazon
`sesarjun@amazon.com`

## Abstract

Recommender systems are ubiquitous in most of our interactions in the current digital world. Whether shopping for clothes, scrolling YouTube for exciting videos, or searching for restaurants in a new city, the recommender systems at the back-end power these services. Most large-scale recommender systems are huge models trained on extensive datasets and are black-boxes to both their developers and end-users [50, 51]. Prior research has shown that providing recommendations along with their reason enhances trust, scrutability, and persuasiveness of the recommender systems [37, 8, 33]. Recent literature in explainability has been inundated with works proposing several algorithms to this end [50]. Most of these works provide item-style explanations, i.e., 'We recommend item A because you bought item B.' We propose a novel approach, `RecXplainer`, to generate more fine-grained explanations based on the user's preference over the attributes of the recommended items [19]. We perform experiments using real-world datasets and demonstrate the efficacy of `RecXplainer` in capturing users' preferences and using them to explain recommendations. We also propose ten new evaluation metrics and compare `RecXplainer` to six baseline methods.

## 1 Introduction

Recommender systems drive the modern discovery of subjects of interest. Examples are ubiquitous, Netflix and YouTube for entertainment, Yelp for restaurants, Amazon and Shein for fashion. Content-based and collaborative filtering (CF) are the two primary approaches of generating recommendations [3, 9]. Content-based recommender systems use the similarity between an item's attributes and a user's preference over them to recommend new items, for example recommending *Top Gun* to a user who likes *Action* movies [11, 4]. Collaborative filtering (CF) methods instead rely on the wisdom of the crowd to generate recommendations. Such methods are trained only using the ratings provided by the users of a recommendation platform like Netflix. The idea is that if two users have a similar rating pattern, then the items liked by a user can be recommended to the other (if they have not already interacted with it) [30, 34]. Hybrid recommender systems aim at combining the two approaches [10].

In recent years, CF has been widely chosen over content-based methods owing to 1) ease of training – labeling costs of new items is high as their attributes need to manually labeled [20], and 2) better recommendations in terms of serendipity and discovery [17]. However, this comes with one major limitation. Content-based methods are transparent and scrutable as they generate recommendations using a user's attribute preference, but this is not the case with CF-based methods. CF methods

2022 Trustworthy Embodied AI (TEA 2022) co-located with NeurIPS 2022.

map users and items to an embedding space, which is learned from the user-item interaction matrix (consisting of all users and all items) – and the proximity in this space is used to generate recommendations. Such an embedding space is difficult to interpret. A core challenge is understanding what a model learns about the user's preference over the items and explaining how it generates the recommendations.

Previous research has established that providing reasons for a recommendation enhances the transparency, scrutability, trustworthiness, effectiveness, persuasiveness, efficiency, and satisfaction of the recommender systems [37, 8, 33]. Providing explanations when using CF-based methods is nontrivial. This has spurred significant research in the broad field of "explainability for CF recommender systems". Most of the previous explanation generating approaches provide explanations in the form of either user-based or item-based explanations. User-based explanations explain a recommendation on the basis of 'similar' users liking it. And item-based explanations explain a recommendation on the basis of its closeness to other items that the user has liked in the past. Item-based explanations are usually easier to grasp as the user knows about the items they interacted with in the past. However, both of these explanation formats do not capture a user's preference over the attributes of an item — which is how users inherently think about a recommendation [19, 42, 16, 23, 24, 43]. (Providing attribute-based explanations is much easier when using content-based recommender systems, as users' preference is what those systems directly use to generate recommendations.)

This work proposes a novel approach, `RecXplainer`, that generates attribute-based explanations for CF-based recommender systems. A recommendation is explained in terms of a user's preference over the attributes of this item, e.g., 'We are recommending you this movie because you like *Action* movies.' Such explanations are personalized to the user and hence help further enhance the persuasiveness and trustworthiness of the recommender systems.

Previous research has also established the significance of post-hoc and model-agnostic explainability, citing two reasons: 1) to evade the accuracy-interpretability trade-off (simpler and interpretable models are usually less accurate), and 2) generalizability to all recommender systems [43, 28, 25]. Aligning with this school of thought, our proposed approach is post-hoc and model-agnostic – it can generate explanations for any recommender system that operates using user and item embeddings.

We found this important intersection under-researched in the literature. We could only find two methods that generate attribute-based explanations, LIME-RS [26] and AMCF [27]. LIME-RS did not evaluate its attribute-based explanations, and the metrics used by AMCF were not justified and not generalizable (see Appendix A). To this end, we also propose a set of generalizable and well-justified metrics to evaluate attribute-based explainability techniques for recommender systems (Section 4).

In summary, our contributions are:

1. We propose `RecXplainer`, a novel, post-hoc, and model-agnostic technique to provide attribute-based explainability for recommender systems, a largely neglected research area.

2. We propose several metrics to evaluate attribute-based explainability methods for recommender systems that can be used to compare all such future techniques.

3. We explore comparison of `RecXplainer` and other methods with the often overlooked popularity-based methods.

## 2 Related Work

Machine Learning (ML) is being increasingly used to automate decisions. Some of the applications where ML is being used are highly critical and directly affect humans, for example, loan approval [32], criminal justice [36], and hiring [31]. The nascent field of trustworthy ML aims to detect bias in ML models (and counteract it), understand the factors that the ML model is using in making predictions, ensure the models respect privacy and security, and frame policies and regulations that the ML models should abide by [6, 5, 46]. In this work, we focus on explainability and refer the readers to other works for a broad discussion of trustworthy ML [6], [15], [40].

## 2.1 Explainability in Recommender Systems

Explainable recommender systems provide recommendations along with a justification for doing so. The term 'explainable recommendation' was introduced by Zhang et al. [51] in 2014, however there were papers talking about the benefits of providing explanations much earlier [38, 39, 33].

As mentioned, most previous methods for explainability in recommender systems provide explanations in user-based or item-based fashion [50]. The approaches that generate these explanations can either be model-specific or model agnostic, the former constituting the majority of previous methods. Models-specific approaches train inherently interpretable models by constraining the embedding space [28]. Zhang et al. [51] proposed extracting user preferences over attributes from item reviews and using that to provide explanations. They train an explicit factor recommender model that takes the user's preference over attributes as an input during training. Abdollahi and Nasraoui [1] add constraints in the latent space of the matrix factorization models to encourage explainability – they provide user and item-based explanations. There exist many other approaches in this category [13, 12, 7, 35, 21, 23, 49]. A major downside of these approaches is that they are *not* post-hoc, i.e., they train a separate model for each recommendation use case, and this is problematic for two primary reasons: 1) there is a potential accuracy-interpretability trade-off, and 2) industrial teams have fine-tuned their recommender models over large datasets, and are rarely willing to train a new model from scratch in order to provide explanations. Hence post-hoc explainability techniques attract lot of attention, especially from industry perspective [41, 25].

## 2.2 Post-hoc Explainability in Recommender Systems

There only exist a handful of explainability approaches in this category. Peake and Wang [28] use a data-mining approach to provide item-based explanations in a post-hoc manner. Nóbrega and Marinho [26] propose LIME-RS which is motivated from the LIME paper [29]. It samples items close to a recommended item, learns a linear regression model on these samples, and uses the regressor to provide an explanation that can be either item-based or attribute-based (the latter can be provided if the attributes of the items are appended to the items embeddings when training the regression model). Cheng et al. [14] propose a technique that uses influence functions to find the influence of training rating on a particular recommendation, and the ratings with the highest influence are served as explanations. This technique, therefore, provides item-based explanations.

## 2.3 Attribute-Based Explainability in Recommender Systems

Unlike user- and item-based explanations, attribute-based explanations learn a user's personalized attribute preferences, thereby increasing the system's persuasiveness and trustworthiness [44, 16]. As mentioned earlier, there do exist works that provide explanations which utilize the user's preference over interpretable attributes of an item. Most of these works utilize the reviews provided by the users to capture this preference (and use that to train the recommender system itself [51, 23, 19]). However, such reviews might not exist even in the presence of ratings, for example, Netflix scrapped its review section due to low user participation recently [48]. Secondly, there is no guarantee of mention of interpretable attributes in the reviews that can be used to learn the user's preference. Even the datasets used in these papers had very sparse user reviews; over 77% of the users had only *one review* [18], making the inference of users' preferences questionable.

On the other hand, our approach also provides attribute-based explanations by utilizing the attributes of an item available in most recommender datasets (e.g., movie or game genres), and it *does not* use reviews. Similar to ours, AMCF [27] learns users' preferences over item attributes by training an attention network. So it provides attribute-based explanations like our approach; however, AMCF is not a post-hoc technique. Similarly, Wang et al. [44] proposes CERec to determine the attributes that are important for a user-item pair (in a counterfactual manner); however, CERec is also not a post-hoc technique. Wang et al. [43] propose a post-hoc RL-based approach that generates attribute-based explanations for a user; however, like most previous approaches, it requires access to the user reviews.

Hence our approach sits at the intersection of attribute-based (*without* using user reviews) and post-hoc explanation techniques. To the best of our knowledge, LIME-RS is the only previous approach that falls in this category.

## 3 `RecXplainer`: Algorithm

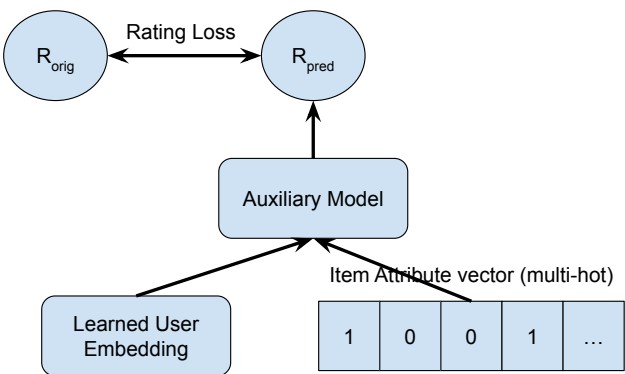

Figure 1: `RecXplainer`'s architecture.

Figure 1 shows the architecture of `RecXplainer`. We train a model (*Auxiliary Model* in the diagram) that takes as input a user's embedding from the trained recommender model and an item's attribute vector. This attribute vector is multi-hot – it has 1s when the item has those attributes, otherwise, 0s (instead of binary values, the vector could also have continuous values). The model is trained for all ratings present in the training dataset, and the goal is to reproduce the rating as given by the user ($R_{orig}$ in the diagram). The model is trained using a mean-squared error between the original rating and the predicted rating ($R_{pred}$ in the diagram). The auxiliary model can be instantiated with any ML model. For experiments, we used three models: Linear, a 2-layer neural network, and a 4-layer neural network. We train the model using Adam and SGD optimizers until convergence.

### 3.1 Generating Explanations

Once the auxiliary model is trained, we use it to generate explanations for recommended items. For a given user-item pair for which an explanation needs to be generated, `RecXplainer` produces a sequence of that item's attributes ranked in decreasing order of that user's preference. We term this a user's *specific preference* over an item's attributes. The preference of a particular attribute (for a given user-item pair) is determined as the loss in predicted rating if that attribute was not there in the item. For example consider a movie who genres are *Crime*, *Documentary*, and *Horror*. For a user Alice, the auxiliary model predicts a rating of 4.2 for this movie. Now, we can use the auxiliary model to predict this movie's rating if the genres are zeroed out one by one. Table 1 shows this procedure. We zero out the attributes present in the item, one by one, and see the drop in rating (last column). The higher the drop, the higher the rank of the removed attribute. Our approach of attributing importance to attributes falls in a well-studied domain of explainability – broadly termed as removal-based explanation methods. We provide more details in Appendix A. In the example, the ranked order of preference is *Crime*, *Horror*, and *Documentary*. This is our analog of *local interpretability* in the context of classification in ML provided by popular methods like LIME [29] and SHAP [22].

Table 1: Illustration of how `RecXplainer` computes the specific preference of a user over an item's attributes (in this case, genres of a movie).

| Attribute zeroed | Predicted rating | $\Delta$(Predicted rating) |
|---|---|---|
| No attribute zeroed | 4.2 | – |
| *Horror* | 3.7 | 0.5 |
| *Documentary* | 3.9 | 0.3 |
| *Crime* | 3.2 | 1.0 |

`RecXplainer` also generates the *general preference* of a user over all attributes in a dataset by averaging the specific preferences over all the items that a user 'liked' in the past (note that this is not the same as all items that a user rated in the past). This is our analog of *global interpretability* in the context of classification in ML.

# 4 Evaluation

Our experiments characterize the quality of explanations for every user by the number of liked items in the user history and the number of top-$k$ recommendations that can be explained.

## 4.1 Experimental Methodology

**Metrics.** As mentioned in Section 2, there are only two previous methods that can generate attribute-based explanations for recommender systems: LIME-RS [26] and AMCF [27]. However, LIME-RS did not quantitatively evaluate its attribute-based explanations, and the AMCF did not justify its metric of choice (see Appendix A). Hence, we propose a set of 10 generalized metrics to evaluate any attribute-based explanation method for recommender systems. We evaluate both general and specific preferences in these metrics.

1. *Test set coverage:* This metric finds if the general preferences of a user identified by a technique have any intersection with the attributes of the items in the test set that they liked. For example, suppose the identified general preferences for a user Adam are *Anime*, *Comedy*, and *Sci-Fi* and there is a movie that Adam liked whose genres are *Action* and *Anime*. In that case, we could count this movie as covered. Had the genre of the movie only been *Action*, then it would not have been covered. For Movielens-$100K$, we consider a movie 'liked' if a user has rated it $4$ or $5$. We only consider the top-3 general preferences for a user when measuring coverage. We report the mean coverage over all the users for this metric.

2. *Top-$k$ recommendations coverage:* This metric finds if the top-3 identified general preferences covers (i.e., have any intersection) with the attributes of the top-20 recommended items to a user. We report the mean coverage over all the users for this metric.

3. *Personalization of the explanations:* Since a few attributes in most recommender datasets are very popular, i.e., they occur in almost all items: identifying such an attribute as a user's preference will provide almost 100% coverage for the metrics mentioned above – however, this might be an inaccurate and unpersonalized explanation. To measure how personalized the explanations are, we need to know the ground-truth users' preferences over the set of attributes – unfortunately, this is unknown to us. To overcome this limitation, we propose two methods that act as reasonable proxies for a user's attribute preference (The gold standard would be to conduct a user study; however, that would involve collecting a new recommender dataset and asking users to tell us about the preferences, which we leave to future work.):

   (a) *Conditional Probability of Liking given a genre is present:* This measures the probability that a user likes a movie, given that the attribute is present in a movie. It is the ratio of the number of times a genre is present in the movies a user likes (rated 4 or 5) and the number of times it is present in all the movies the user rated (rated 1 through 5).

   (b) *Odds of Liking vs. Disliking: given a genre is present:* This measures the ratio of the number of times a genre is present in a movie a user likes (rated 4 or 5) and the number of times it is present in a movie that the user dislikes (rated 1 or 2).
   We use the training set for calculating both these preference proxies.

   Both provide a reasonable proxy of users' general preference over genres, i.e., weights over the set of genres. We report four metrics considering such weights as ground truth user preference:

   (a) *General preferences coverage:* This metric measures if there is any intersection between the top-$k$ general preferences identified for a user and the top-$k$ preferences identified by either of the two proxies mentioned above.

   (b) *General preferences ranking:* This metric measures the similarity between the ranking of the top-$k$ general genre preferences identified for a user and the top-$k$ preferences identified by either of the two proxies. We use rank-biased overlap (RBO) as a measure of similarity between the two ranked lists (we justify this choice in Appendix A).

   (c) *Specific preferences coverage:* Similar to general preferences, we also measure if there is any intersection between the top-$k$ genre preferences for each item that a user liked in the training set (specific genre preference for a user-item pair) and the top-$k$ genre preferences identified for this user by either of the two proxies.

(d) *Specific preferences ranking:* We also measure the similarity in the ranking of the top-$k$ genre preferences for items liked by a user in the training set and the top-$k$ genre preferences identified for this user by either of the two proxies.

For all the above metrics, we report the mean over all the users. Hence we have a total of 8 metrics for measuring the personalization of the explanations. We used k = 3 in the experiments.

**Dataset.** We used the popular recommender dataset Movielens-$100K$ for our experiments. Movielens-$100K$ has 100,000 ratings for 1682 movies provided by 943 users. Each rating is an integer from 1 to 5, with a higher number meaning a higher likeness of it. The dataset also contains the genre of each movie which is a multi-hot vector of size 18: there are a total of 18 genres in the dataset: *Action, Adventure, Animation, Children's, Comedy, Crime, Documentary, Drama, Fantasy, Film-Noir, Horror, Musical, Mystery, Romance, Sci-Fi, Thriller, War, Western* and each movie is categorized as having one or more genre. These are the attributes that a technique identifies a user's preference over. We used 70% of the dataset for training a matrix factorization-based recommender system and 30% of the data as the test set (data split is stratified by users).

**Baselines.** We compare `RecXplainer` to six baseline methods:

- *LIME-RS [26]:* The only previous post-hoc attribute-based explainability method.
- *AMCF [27]:* Another attribute-based explainability method – this is not a post-hoc method.
- *AMCF-PH:* The AMCF approach adapted for being post-hoc (details in Appendix A).
- *Global popularity:* The genres that are the most popular among all the movies that are liked across the entire dataset. For Movielens-$100K$, these were *Action*, *Comedy*, and *Drama*.
- *User-specific popularity:* The most popular genres for a particular user. These are calculated based on only the movies they liked (rated 4 or higher).
- *Random:* This baseline is for control, for each metric, it selects a random set of genres for each user-item pair, when computing specific preferences, and for each user when computing the general preferences.

Table 2: The test set coverage, top-k recommendations coverage, and the 8 explanation personalization metrics are reported here (averaged over all users). For `RecXplainer`, we computed metrics with three auxiliary models: Linear, a 2-layer neural network (MLP1), and a 4-layer neural network (MLP2). For all columns, a higher value is better. The highest value for each metric is emboldened.

| Metrics | LIME-RS | AMCF | AMCF-PH | GBL Popl. | User Popl. | Random | Our-Linear | Our-MLP1 | Our-MLP2 |
|---|---|---|---|---|---|---|---|---|---|
| Testset Coverage | 8.4 | 47.2 | 52.6 | **84.5** | 77.9 | 32.4 | 60.7 | 70.8 | 71.2 |
| Recommendations Coverage | 26.0 | 44.3 | 46.1 | **73.2** | 69.6 | 30.6 | 61.3 | 68.6 | 67.8 |
| CondProb Generalpref Coverage | 50.3 | 58.2 | 47.2 | 29.0 | 41.8 | 43.9 | **62.9** | 58.1 | 56.6 |
| CondProb Generalpref Ranking | **15.8** | 15.1 | 11.2 | 5.8 | 9.9 | 11.0 | 15.2 | 14.0 | 13.8 |
| CondProb Specificpref Coverage | 52.5 | 53.1 | 52.4 | 29.0 | 41.7 | 44.1 | **55.3** | 53.8 | 53.6 |
| CondProb Specificpref Ranking | 14.6 | 49.6 | 50.3 | 6.8 | 10.4 | 11.0 | **52.2** | 50.7 | 51.2 |
| Odds Generalpref Coverage | 32.6 | 63.7 | 60.4 | 48.5 | 55.0 | 43.7 | **73.4** | 72.3 | 70.4 |
| Odds Generalpref Ranking | 9.4 | 19.4 | 18.0 | 18.2 | 26.7 | 10.6 | 25.6 | 27.6 | **28.2** |
| Odds Specificpref Coverage | 37.2 | 55.8 | 56.8 | 48.5 | 55.0 | 44.2 | **59.8** | 59.2 | 59.5 |
| Odds Specificpref Ranking | 10.2 | 49.7 | 50.7 | 19.6 | 27.2 | 11.0 | 50.8 | 51.6 | **51.7** |

## 4.2 Discussion

Table 2 reports all the metrics for all the baselines and `RecXplainer`. We use it to answer our research questions:

1. *LIME-RS:* The test set and top-$k$ recommendations coverage provided by LIME-RS is abysmally low, even worse than the *random* baseline. It trails `RecXplainer` in all the 10 metrics, barring the ranking of general preferences when the ground truth preference is calculated using the conditional probability of liking.

2. *AMCF:* This approach trails `RecXplainer` in all metrics.

3. *AMCF-PH:* The post-hoc adapted version of AMCF also trails `RecXplainer` in all metrics.

4. *Global popularity:* It achieves the highest test set and top-$k$ recommendations coverage among all techniques, and understandably so — the three most common genres *Action*, *Comedy*, and *Drama*

occur is over 88% of the test set items (see the genre distribution plot in Figure 3). However, it performs much worse than `RecXplainer` on all personalization metrics, many times even worse than the *random* baseline. Therefore, we can conclude that the explanations served by this approach do not capture a user's preference over genres and can perform well for coverage-based metrics because of their skewed distribution.

5. *User-specific popularity:* This popularity approach achieves the second highest test set and top-$k$ recommendations coverage, and similar to *global popularity* performs worse than `RecXplainer` on all personalization metrics, many times even worse than the *random* baseline.

6. *Random:* We used this approach to serve as control and to ensure that no metric was trivial to perform well on. Not surprisingly, it performs worse than `RecXplainer` on all metrics.

7. *Ours:* `RecXplainer` has the third highest test set and top-$k$ recommendations coverage (approaching the performance of user-specific popularity for the latter) while performing the best on all personalization metrics except for the general preferences ranking when using the conditional probability of liking, where it performs the second best. Therefore, we can conclude that `RecXplainer` serves the most personalized explanations while still being able to explain a large proportion of test set items and top-$k$ recommendations.

## 5   Conclusions

As our metrics indicate, `RecXplainer` strikes a great balance between coverage and personalization of the explanations. We would also like to mention that neither of the previous techniques, LIME-RS or AMCF, considered popularity as a baseline. We show that popularity approaches perform pretty well on the two coverage metrics (those are the metrics that were used in their papers). Even the item-based explainability papers [1, 28] did not compare to popularity even though the metrics they considered were precision and recall. Since recommender systems are known to have popularity bias [2, 52], popular items and popular attributes can perform very well on these metrics. Hence we consider discussing popularity as a potential explanation for both item-based and attribute-based explanations as a contribution of our work.

## 6   Limitations

The only limitation of this approach is the cost incurred during training of the auxiliary model. In our experiments, the training of the three model architectures were all very fast, all of them were trained in less than 10 minutes on a laptop.

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

# A  Appendix

**AMCF Evaluation Metrics:**  AMCF also evaluate their general and specific preference. For general preference, similar to our evaluation metrics, their primary metric is *Top M recall at K*, i.e., the intersection between the top M ground truth user preference and top K predicted user preferences. To realize this metric, they need the ground truth user preference – which is not present in the dataset – and hence similar to our technique, they need to simulate. The way they compute the ground truth preference is unjustified and inexplicable. The process involves computing the weight of each item by removing the user and item bias terms, and then the preference of an attribute is just the sum of the weights of the items it occurs in. The latter part is still reasonable; however, we are uncertain about the weight calculation part. For evaluating specific preferences, they use the sorted order of general preference for the attributes present in that specific item – which does not resolve the concerns mentioned above.

We also use proxies to simulate the users' ground truth attribute preference; however there are key differences in our evaluation when compared to AMCF:

- The proxies are more generalizable and reasonable, like the conditional probability of liking and odds of liking.
- We use multiple proxies and report results on all of them to avoid cherry-picking.

**Justification of the removal-based explanation:**  Covert et al. [15] developed a framework to categorize such methods along three dimensions:

- Attribute removal: how the approach removes attributes from the model,
- Model behavior: what model behavior is it observing, and
- Summary technique: how does it summarize an attribute's impact?

`RecXplainer`, when instantiated in this framework: removes attributes by setting them to zero, analyzes prediction as the model behavior, and summarizes an attribute's impact by removing them individually.

Since our attribute input vector is binary and indicates whether an attribute is present or not, simulating an attribute's removal by setting it to zero is a natural choice. Previous removal-based explanation methods have used *prediction* or *prediction loss* or *dataset loss* for model behavior analysis. We chose *prediction* instead of *prediction loss* for our analysis because we wanted to get specific preference (analog of local interpretability) without requiring the original rating. Previous removal-based explanation methods have used *removing individual attributes* or *Shapley values* or *trained additive models* to get attribute impact value. Removing individual attributes accesses the impact of an attribute by measuring the loss in prediction when that one attribute is removed, and this is what we choose. On the other hand, Shapley values take all subsets of attributes and then use the cooperative game theoretic formulation to assign impact value to each attribute. It has two disadvantages:

- It creates all subsets of attributes – which is exponential in the number of attributes, making the process very expensive.

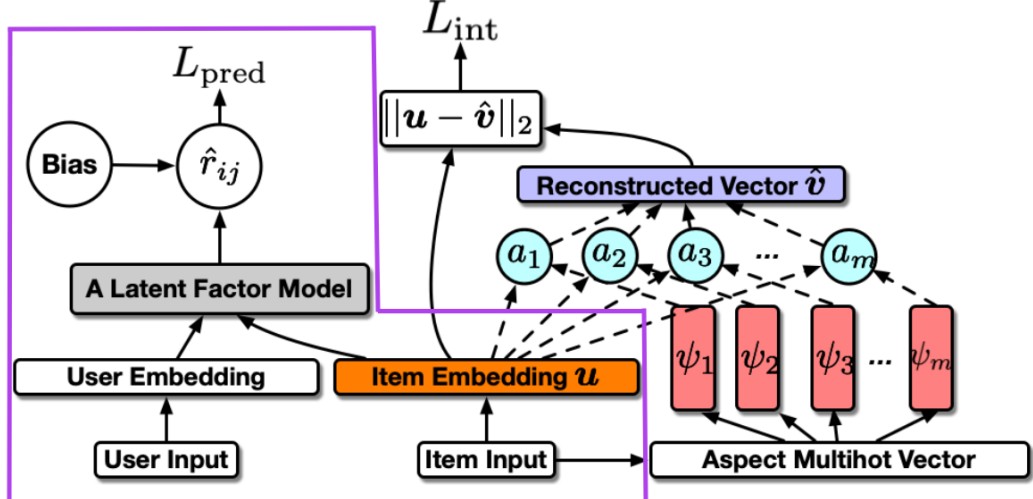

Figure 2: The architecture of AMCF Post-Hoc (AMCF-PH). The part within the blue colored region was not trained, thereby making the technique post-hoc.

- For creating all the subsets, it simulates removing many features that can potentially create attribute vectors that the auxiliary model has not seen, and thereby its prediction can not be trusted in that part of the data manifold.

For these reasons, we choose the method of removing individual attributes. There are potential downsides to this choice as well. If the ground truth reason for recommending a movie is either if it were either *Horror* or *Crime*, zero-ing out one of them will not make a difference in prediction, and hence their assigned impact will not be correct.

**Justification of using Rank-Biased Overlap (RBO):**

Comparing ranked lists can be achieved using various rank correlation metrics like Kendall's Tau and Spearman's correlation [47]. These metrics have restrictions that the ranked lists must be conjoint, i.e., they both must contain all the items that the entire universe of items contains. This is not suitable for our metrics because the top-$k$ attributes from either the conditional probability or the odds liking proxy might not have any common element with the ranking produced by any technique. Hence we choose to compare the ranked lists using rank-biased overlap (RBO) metric [45], which was recently proposed to overcome the limitation posed by correlation-based metrics. Specifically, RBO is better than correlation-based metrics as:

- RBO does not require the two ranked lists to be conjoint, i.e., a ranked list having items that do not occur in the other list is acceptable, e.g., the similarity between [1, 3, 7] and [1, 5, 8] can be measured using RBO.
- RBO does not require the two ranked lists to be of the same length.
- RBO provides weighted comparison, i.e., discordance at higher ranks is penalized more than discordance at lower ranks.

**AMCF Post-Hoc Adaptation:**

For adapting AMCF [27] to be post-hoc, we made a minor modification to the architecture mentioned in their paper (see Figure 2). We froze the left-hand side of the architecture and only trained the right-hand side, which consists of the attention network that aims to reconstruct the item's embedding from its attributes. Since the user and item embeddings were not trained, this made the technique post-hoc.

**Figure 3 shows the genre distribution for Movielens-100K.**

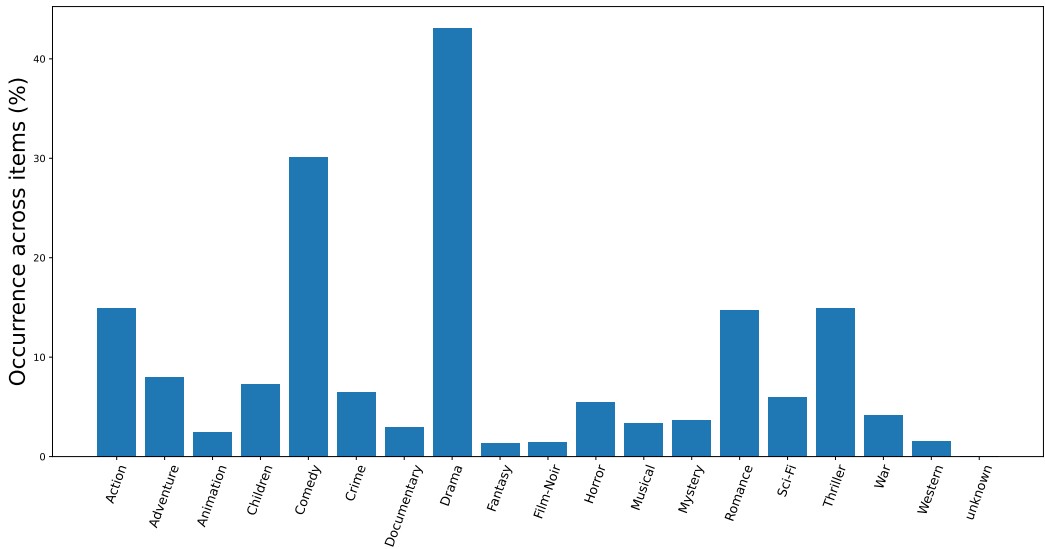

Figure 3: The genre distribution for Movielens-100K dataset. The top-3 most popular genres: *Action*, *Comedy*, and *Drama* occur in over 88% of the movies.

