# OpenReview forum: "RecXplainer: Post-Hoc Attribute-Based Explanations for Recommender Systems"
_NeurIPS.cc/2022/Workshop/TEA — TEA_

### Official Review · Reviewer_2ru3 · 2022-10-16
**Good contribution for TEA**

**Rating:** 7
**Confidence:** 4

**Review:**

Summary
This paper notes that common explanations of the form "We recommend A because you purchased B" are not sufficiently fine-grained, as they do not reflect user's preferences over the input attributes for recommender systems. To address this, the paper proposes a new method for generating explanations.

Strengths
This paper is thorough in its introduction of and analysis of a new method. It is well-written (especially the first half) and well motivated.

Weaknesses
I would like to see a discussion of limitations added to the paper. E.g., what is the cost of training an auxiliary model?

The phrasing of the research questions is a bit convoluted and could be simplified, e.g., "How many of the test-set items that a user liked can the general preferences identified by an
approach explain?"

Other comments
Should be "Top Gun" not "Top Guns" ;)

Pt 2 in the "In summary, our contributions are" is not really a contribution. These are attributes of the method.

I think the accuracy/interpretability trade-off is quite contested. I'm not exactly sure where current discourse stands, but drawing this to the attention of the authors for future submissions.

---

### Official Review · Reviewer_wpxD · 2022-10-18

**Rating:** 8
**Confidence:** 3

**Review:**

This paper addresses the challenge of producing post-hoc explanations for recommendations made using collaborative filtering methods. The contributions of this work include an algorithm and model for producing attribute-based explanations and a set of evaluation metrics for comparing recommendation explanations.

Overall, the paper is well-written and clearly motivates the proposed method. The approach itself is clear, and the evaluation metrics themselves are a useful contribution that I expect will result in interesting discussion at the workshop. The proposed algorithm produces three explanation models, which collectively result in high performance on the evaluation metrics. Interestingly, the simplest model (linear) resulted in the best performance, which I would have liked to see discussed more.

---

### Official Review · Reviewer_m9et · 2022-10-20
**Very Interesting and Important Topic**

**Rating:** 8
**Confidence:** 4

**Review:**

This paper is one of the first to study the attribute-based recommender system, a largely overlooked problem. This problem also has great practical significance. The authors developed a novel method that is post-hoc and hence model-agnostic, thus can be applied to any recommender system. The authors also considered how to formally evaluate attribute-based explainability methods for recommender systems, and proposed principled metrics for that. This paper addressed a very important problem, proposed novel approaches, and has a comprehensive experimental analysis. I think it can make a good contribution to the workshop.

---

### Decision · Program_Chairs · 2022-10-21

**Decision:**

Accept

**Comment:**

All the reviewers reach an agreement that the paper is well-written and well-motivated. Please consider incorporating the comments from the reviewers, especially Reviewer 2ru3, into the final version.